# Exploring the contributions of vegetation and dune size to early dune development using unmanned aerial vehicle (UAV)-imaging

Short running head: Dune size and vegetation

Marinka E.B. van Puijenbroek[1], Corjan Nolet[2], Alma V. de Groot[3], Juha M. Suomalainen[4,5], Michel J.P.M. Riksen[2], Frank Berendse[1] and Juul Limpens[1]

[1]Plant Ecology and Nature Conservation Group (PEN), Wageningen University & Research Wageningen, P.O. Box 47, 6700 AA, The Netherlands

[2]Soil Physics and Land Management Group, Wageningen University & Research, Wageningen, P.O. Box 47, 6700 AA, The Netherlands

[3]Wageningen Marine Research, Wageningen University & Research, Den Helder, Ankerpark 27, 1781 AG, The Netherlands

[4]Laboratory of Geo-Information and Remote Sensing, Wageningen University & Research, Wageningen, P.O. Box 47, 6700 AA, The Netherlands

[5]Finnish Geospatial Research Institute, National Land Survey of Finland, Kirkkonummi, Finland

*Correspondence to*: Marinka E.B. van Puijenbroek (marinka.vanpuijenbroek@gmail.com)

**Abstract**

Dune development along highly dynamic land-sea boundaries is the results of interaction between vegetation and dune size with sedimentation and erosion processes. Disentangling the contribution of vegetation characteristics from that of dune size would improve predictions of nebkha dune development under a changing climate, but has proven difficult due to scarcity of spatially continuous monitoring data.

This study explored the contributions of vegetation and dune size to dune development for locations differing in shelter from the sea. We monitored a natural nebkha dune field of 8 hectares, along the coast of the island Texel, the Netherlands, for one year using an Unmanned Aerial Vehicle (UAV) with camera. After constructing a Digital Surface Model and orthomosaic we derived for each dune 1) vegetation characteristics (species composition, vegetation density, and maximum vegetation height), 2) dune size (dune volume, area, and maximum height), 3) degree of shelter (proximity to other nebkha dunes and the sheltering by the foredune). Changes in dune volume over summer and winter were related to vegetation, dune size and degree of shelter.

We found that a positive change in dune volume (dune growth) was linearly related to initial dune volume over summer but not over winter. Big dunes accumulated more sand than small dunes due to their larger surface area. Exposed dunes increased more in volume (0.81% per dune per week) than sheltered dunes (0.2% per dune per week) over summer, while the opposite occurred over winter. Vegetation characteristics did not significantly affect dune growth in summer, but did significantly affect dune growth in winter. Over winter, dunes dominated by *Ammophila arenaria*, a grass species with high vegetation density throughout the year, increased more in volume than dunes dominated by

*Elytrigia juncea*, a grass species with lower vegetation density (0.43 vs. 0.42 $(m^3/m^3)$/week). The effect
of species was irrespective of dune size or distance to the sea.
Our results show that dune growth in summer is mainly determined by dune size, whereas in winter
dune growth was determined by vegetation. In our study area the growth of exposed dunes was likely
restricted by storm erosion, whereas growth of sheltered dunes was restricted by sand supply. Our
results can be used to improve models predicting coastal dune development.
**Key words**: Nebkha dunes, *Ammophila arenaria*, *Elytrigia juncea*, beach-dune interaction, landform
morphology, the Netherlands

## 1. Introduction

Coastal dunes occur along the sandy shores of most continents (Martínez and Psuty, 2008), and are important to protect these coasts against flooding, provide areas for recreation, store drinking water and shelter unique biodiversity (Everard et al., 2010). Coastal dunes and their services are threatened by climate-induced sea-level rise (Carter, 1991; Feagin et al., 2005; Keijsers et al., 2016). However, dunes also provide self-adapting systems of coastal protection, since the threat by sea-level rise can be mitigated by the development of new dunes. Although the development of new dunes is well described, we know little about the factors that determine the speed of early dune development. Understanding these factors is essential for predicting dune development, and for safeguarding their services.

Dune development is the result of an interaction between vegetation and aeolian processes and starts above the high-water line by the establishment of dune-building plant species (Maun, 2009). Once vegetation establishes on the bare beach, it forms a roughness element that facilitates local sand deposition and reduces erosion, forming a small dune within discrete clumps of vegetation (Dong et al., 2008; Hesp, 2002). At the lee side of these small clumps of vegetation a shadow dune develops by sand deposition, this shadow dune has a ridge parallel to the wind direction (Clemmensen, 1986; Gunatilaka and Mwango, 1989; Hesp, 1981). Vegetation and shadow dune together are known as nebkha dunes, embryo dunes or incipient foredunes (Hesp, 2002; Hesp and Smyth, 2017). The further development of these nebkha dunes strongly depends on the balance between summer accumulation of sand and vegetation growth and winter erosion of sand and loss of vegetation (Montreuil et al., 2013). Summer growth and winter erosion depend on weather conditions, such as wind speed, precipitation and storm

intensity (Montreuil et al., 2013; van Puijenbroek et al., 2017). As a result, net dune growth can differ
from year to year. Over time the smaller vegetated dunes can develop into an established foredune that
forms the first line of coastal defense against flooding.

73        Most research on coastal dune growth and erosion have focussed on processes and factors that

influence the supply of sand to the dunes and the effect of storm intensity on dune erosion (Anthony,
2013; Haerens et al., 2012; Houser et al., 2008; Keijsers et al., 2014; Saye et al., 2005; de Vries et al.,
2012). However, how coastal nebkha dune growth and erosion rates are influenced by the individual
dune characteristics, such as dune size, vegetation and degree of sheltering are less well studied. Dune
size affects the wind flow pattern, thus affecting sand deposition (Walker and Nickling, 2002) for
example increasing height or length of the shadow dune (Hesp, 1981; Hesp and Smyth, 2017). Dune
size also influences storm erosion: Claudino-Sales (2008) found that foredunes with a higher volume
were less sensitive to erosion. Whether the latter also applies to nebkha dunes, is unknown. Differences
in vegetation density between plant species are known to modify sand deposition (Arens, 1996; Hesp,
1983; Keijsers et al., 2014; Zarnetske et al., 2012), storm erosion (Charbonneau et al., 2017; Seabloom
et al., 2013), and dune morphology (Du et al., 2010; Hacker et al., 2012; Hesp, 1988). Sheltering by
other nebkha dunes can decrease the sand supply but can also reduce erosion by waves (Arens, 1996;
Lima et al., 2015; Luo et al., 2014; Montreuil et al., 2013). Although dune size, vegetation and
sheltering are known to be important for individual nebkha dune development, the relative contributions
of these factors are unknown.

In this study, we explored the contribution of vegetation and dune size to dune development. Using an unmanned aerial vehicle (UAV) with camera we monitored a natural nebkha dune field for one year. From the aerial images we constructed digital terrain models (DTM) and orthomosaics. From the DTM's and orthomosaics we extracted detailed data on dune size (dune area, volume and maximum height), vegetation characteristics and the degree of sheltering. We related changes in dune volume (dune growth) to initial dune size, vegetation and sheltering over a summer (April - August) and winter period (November - April). We expected that nebkha dune growth would be a function of vegetation density, initial dune size, and shelter, with the function being modulated by season and degree of shelter. We hypothesised that:

1. Nebkha dunes with high vegetation density grow faster irrespective of season or shelter.
2. In summer, growth of nebkha dunes is linearly related to initial dune size, with small dunes growing at the same rate as big dunes. Exposed dunes grow faster than sheltered dunes because of higher sand supply.
3. In winter dune growth is no longer linearly related to initial dune size, as small dunes are more susceptible to storm erosion than big dunes. Exposed dunes grow slower than sheltered dunes because of higher storm erosion.

## 2. Methods

### 2.1 Study site

We monitored 8 hectares (200 m x 400 m) of a natural nebkha dune field with a large range of dune sizes at 'the Hors', the southern tip of the barrier island at Texel, the Netherlands, coordinates: 52°59'43.70"N, 4°43'47.53"E (Fig. 1). The Hors is a wide dissipative beach with a high degree of hydrodynamic reworking of the sand, which results in a high transport potential and opportunity for dunes to develop. In the last 5 years, between 2010 and 2015, many nebkha dunes have developed on the beach by plant species *Ammophila arenaria*, *Elytrigia juncea* or a mixture of both species. These three dunes with different species composition occur at similar distances from the sea, making this area ideal for exploring the effects of dune size and species composition on dune growth. *A. arenaria* and *E. juncea* differ in their vegetation characteristics: *A. arenaria* grows in dense patches, whereas *E. juncea* has a more sparse growth form. This difference in growth form probably also results into a different dune morphology: *A. arenaria* forms higher 'hummocky' shaped dunes, whereas *E. juncea* builds broader and lower dunes (Bakker, 1976; Hacker et al., 2012). The monitoring area is bisected by a low (maximum height of 7 m NAP, i.e. above the mean sea level near Amsterdam), continuous foredune ridge that runs parallel to the shore. The nebkha dunes that occur at the seaward side of this foredune are more exposed to the sea, while the nebkha dunes occurring at the landward side of the foredune are more sheltered from the sea, enabling us to explore whether the effects of dune size and vegetation are modified by the degree of shelter, especially since the age difference between the seaward and landward nebkha dunes is at most 5 years.

# Figure 1 approximately here #

## 2.2 Weather conditions

Summer conditions during our study period were similar to previous years, while winter conditions
were calmer than usual (Supplementary S1). The precipitation during the growing season was 276 mm,
and the average temperature in June and July was 16 °C. The most common wind direction was South
to South-West. The most common wind speed in summer was 4 - 5 m s$^{-1}$, and the maximum wind speed
was 13 m s$^{-1}$.  In winter the wind speed was higher compared to summer, the most common wind speed
was 5 – 6 m s$^{-1}$ and the maximum wind speed was 19 m s$^{-1}$. We registered one storm during the study
period. This storm, however, could be classified as relatively weak.  The highest water level was 211
cm NAP; compared to 248 cm NAP and 254 cm NAP from previous years. The storm, which was the
first of the season, occurred after the beginning of our mapping campaign.

## 2.3 Data collection

Three UAV flights in November (2015), April (2015) and August (2016) were carried out with a rotary
octocopter UAV system (Aerialtronics Altura Pro AT8 v1) and camera equipment of WageningenUR
*Unmanned Aerial Remote Sensing Facility* (Fig. 1). The octocopter was equipped with a Canon EOS
700D single-lens reflex camera with a 28mm f/2.8 Voigtländer Color Scopar SL-II N objective. The
camera sensor was modified to give a false colour output. The red channel of the camera had been
converted to be sensitive in the near-infrared, with centre point around 720nm. The blue channel of the
camera had been extended to also cover the UV region of the spectrum. The green channel was left with
almost original response. The false colour modification enabled the calculation of a modified

Normalised Difference Vegetation Index (NDVI), a commonly used measure for vitality and/or cover of

the vegetation (Carlson and Ripley, 1997). Aerial images were acquired by auto-piloted flights at an

altitude of 80 m at $4 - 5$ m s$^{-1}$ velocity. The camera was set to take one image per second. The auto-

piloted flights enabled us to have the same flight paths for each of the three mapping campaigns. The

flight paths ensured that images had a minimum of 85% forward and 65% side-way overlap. Four

flights of 10 minutes were needed to cover the study area, yielding up to 900 RAW false colour images

per mapping campaign. Five ground control points were permanently placed in the flight area and

measured with a RTK-DGPS Trimble R6 Model 3 (TSC3) to calibrate our images with coordinates.

During our mapping campaign, a Spectralon reference panel was measured with our camera

immediately before take-off and after landing.

**2.4 Radiometric calibration**

In order to compare the images over the time, they were calibrated and converted from RAW to 16 bit

tiff format. First, we ensured that each individual pixel within an image was comparable, by converting

the RAW digital number into radiance units using a pixel-wise dark current and flat field calibration.

Second, each radiance image was calibrated to a reflectance factor image in order to correct for changes

in incident irradiance on different flight days. This calibration was done by using a Spectralon panel

with a known reflectance factor. The radiometric calibration is described in more detail by Suomalainen

et al. (2014).

The images were subsequently converted into NDVI images. Usage of the standard NDVI was

not possible due to lack of red channel in the false colour modified camera. Thus we used a custom

NDVI equation (Eq. 1), which was recommended by the company that modified the sensor. On their
website (MaxMax.com) this equation was shown to be just as effective for green vegetation as the
traditional NDVI formula ($R^2 = 0.77$) where the red band is taken as the absorption channel.
1) $$NDVI = \frac{(NIR + G) - (2B)}{(NIR + G) + (2B)}$$
Where NIR, G, and B are the near-infrared, green and blue bands of the false colour image respectively.
For photogrammetric reconstruction, the NDVI image layer was stacked with the original green and
blue bands to form a three-color image.

## 2.5 Photogrammetric reconstruction

The large overlap between the consecutive images was necessary for photogrammetric software to
successfully process the aerial images into a 3D point cloud (Fig. 2). The 3D point cloud was generated
using Agisoft Photoscan Professional (v. 1.2.6), using the Structure-from-Motion (SfM) and Multi-
View Stereo (MVS) algorithms (Fonstad et al., 2013; Westoby et al., 2012). The correlated 3D points
are georeferenced to match the ground control points, and contain pixel intensity values of the input
imagery. From this 3D point cloud we interpolated a 5 cm pixel size digital surface model (DSM) and a
1 cm pixel size orthomosaic image. The DSM included also vegetation, which resulted in a vertical
error in dune height in areas where vegetation is present. We removed the vegetation from the point
cloud by identifying and removing the vegetation points. Vegetation points were removed by
distinguishing vegetation from sand using k-means clustering of the 3-D point cloud with NDVI using
the Hartigan and Wong (1979) algorithm in R (R Core Team, 2016). The holes in the point cloud that
arose by removing the vegetation were filled by using LAStools (the tool Blast2dem) (Isenburg, 2016),
which resulted in a Digital Terrain Model (DTM) without vegetation.
# Figure 2 approximately here #
We checked the accuracy of the photogrammetric reconstruction by measuring the vertical error,
the repeatability of the method and the degree in which NDVI predicted the biomass of the vegetation.
The vertical error of the DTM was assessed during a combined mapping and flight campaign in August
2015 by measuring the elevation for 1100 points distributed over the flight area with an RTK-DGPS
Trimble R6 Model 3 (TSC3) and comparing the measured point measurements with the DTM. The
repeatability of the UAV photogrammetry was tested by repeating the same flight path five times in
November 2015 and comparing the similarity between the five DSMs. The NDVI measurements were
tested by clipping the vegetation flush with the sand surface for six *A. arenaria* and seven *E. juncea*
dunes and relating the biomass of the vegetation to the NDVI values.

**2.6 Defining dunes**

To be able to relate dune growth to characteristics of an individual nebkha dune including its shadow
dune, we first had to define individual dunes from the DTM. We followed a step-wise procedure for
each of our mapping campaigns (November, April, and August) using ArcGIS 10.3 (ESRI, 2016) that
resulted into different polygons in which each individual dune expanded or decreased in volume over
the study period. Dune volume and growth were later calculated using the same polygons for each
measurement campaign through time (see next section). To define the polygons we used the step-wise
procedure described below: 1) we constructed a baseline raster by calculating the average elevation in a
circle of 5m radius around each pixel in the DTM. A higher or lower radius resulted in either a too low
or too high baseline. 2) We then qualified pixels of the DTM as dunes, if they were 5 cm or higher
above a baseline raster, or had a slope of 15° or higher. The 5 cm threshold is the minimum that can be
accurately derived from the images and corresponds with visual estimates of nebkha dune foot; a slope
of 15° has been earlier identified by Baas et al (2002), as the slope for a shadow dune. From these
selected 'dune' pixels we created dune polygons. 3) Dune polygons of consecutive campaigns were
overlaid to construct the largest dune-covered area during the study period. 4) Each polygon was
visually checked for minimum size and presence of vegetation: dunes consisting of only one clump of
vegetation (0.4 $m^2$ or smaller) and dunes with no vegetation were discarded to derive conservative
estimates of nebkha dune volume and growth.

**2.7 Variables**
For each nebkha dune and for each mapping campaign we extracted dune volume ($m^3$), max height (m)
and horizontal area ($m^2$) from the dune polygons (see previous section) in the DTM. We calculated
changes in dune volume, i.e. absolute dune growth ($m^3$/week) by subtracting the current dune volume
($V_t$) from the volume of the previous mapping campaign ($V_{t-1}$), correcting for the number of weeks

between the mapping campaigns. To explore relationships irrespective of dune size, we also calculated

the relative dune growth ($m^3/m^3$/week).

We manually identified the species composition on each nebkha dune from the orthomosaic.

Species identification was verified in the field for a random subset of 100 dunes (23%) in May 2016. To

this end we created 2 transects from the southwest border to the northeast border of the area, along

which we determined the species on each nebkha dune. We compared the presence of species in the

field with the orthomosaic, and adjusted the species composition if necessary. In our dataset, dunes have

either *A. arenaria, E. juncea* vegetation, or a mixture of both species. A dune was defined as covered by

a mixture of both species, when it had distinct vegetation patches of both species present. For each

nebkha dune and mapping campaign we also extracted the vegetation density and the maximum plant

height. To assess vegetation density we first distinguished vegetated pixels from non-vegetated pixels

based on the orthomosaic using k-means classification of the NDVI using the MacQueen (1967)

algorithm. Hereafter, the vegetation area ($m^2$) and vegetation density (NDVI/$cm^2$ dune) were calculated

by summing the NDVI values of all vegetated pixels within the dune polygon (vegetation area) and then

dividing this summed NDVI by the total number of $cm^2$ pixels within the dune polygon (vegetation

density). The maximum plant height was calculated by subtracting the DSM (with vegetation) from the

DTM (without vegetation).

Sheltering can affect the sand supply and storm erosion. We used two methods to define the

degree of sheltering. Firstly, we distinguished whether a nebkha dune was seaward or landward from

the foredune. Secondly we determined how much the dune was clustered with other dunes. We

extracted the degree of clustering for each dune by calculating the mean height from the DTM in a 25 m
radius around the dune. All data extraction from the DSM, DTM and orthomosaic were done in R (R
Core Team, 2016).

### 2.8 Statistical analysis

First we explored if nebkha dune area, volume, maximum height, clustering (mean height in a 25m
radius around the dune), vegetation density and maximum plant height depended on species
composition using August 2016 data. As the number of dunes per species composition was unequal, we
used an ANOVA type III SS, to compensate for the unequal sample size (Fox and Weisberg, 2011) and
then used a Tukey HSD test (Hothorn et al., 2008) to determine significant differences between the
dunes with different species compositions.
Secondly, we tested how absolute changes in dune volume over winter (November – April) and
summer (April – August) periods related to the dune volume at the beginning of the period for locations
with different degree in sheltering with a linear regression model.
Thirdly, we analysed how the relative changes in dune volume over winter and summer
depended on dune size and vegetation characteristics in separated linear mixed models (Pinheiro et al.,
2016). To correct for spatial autocorrelation and species distribution we ran this analyses on a subset of
236 (54%) dunes. To this end we first explored the degree of spatial autocorrelation in our dataset by
creating a variogram. To account for the spatial autocorrelation of 25 m in our dataset we imposed a 50

m x 50 m grid over our study area; all dunes that were located within a grid cell (referred to as block) were assumed to show spatial autocorrelation to some extent. This spatial autocorrelation was corrected for in our statistical model by including block as a random intercept. We had 10 blocks seaward from the foredune and 11 blocks landward from the foredune (Fig. 3), in which all species combinations occurred (*A. arenaria* dunes, *E. juncea* dunes and *A. arenaria* + *E. juncea* dunes). By only including dunes that were located within a block in the analysis, our selection was biased towards smaller dunes, since larger dunes often fell within multiple blocks. We do expect that the effect of vegetation is more apparent for these smaller dunes compared to larger dunes. To better distinguish between effects of species compositions and vegetation structure we used two different models. The effect of species composition was tested in a model with dune volume, maximum dune height, clustering and species, whereas the effect of vegetation structure was tested in a model with dune volume, maximum dune height, dune clustering, vegetation density and maximum plant height as explanatory variables. Within each model we used the initial conditions for the explanatory variables, with initial conditions being the values at the start of each measurement campaign. We included all two-way interactions. We selected the best model by using Akaike information criterion (AIC). As we were mainly interested in the importance of the explanatory variables relative to each other, we calculated the standardised estimates for all the models by scaling the explanatory data.

The normality and homogeneity of the variance of the data was visually checked. All statistical analyses were conducted in R (R Core Team, 2016). In the results we use statistic notation to show the results of the ANOVA and linear regression models. We mention the F- value (ANOVA) or t-value (linear regression), which indicates the difference of the explanatory variable to the variation in the data.

The p-value indicates the probability that the null-hypothesis is correct, we used a p-value of 0.05 as a
cut off to reject the null-hypothesis. The number in subscript indicates the degrees of freedom.

**3. Results**
**3.1 Nebkha dune characteristics**
Within the 8 hectare nebkha dune field we distinguished 432 polygons that were covered with nebkha
dunes for at least one moment during our mapping campaigns (Supplementary material S2). Half of
these dunes were covered by *E. juncea* vegetation (50.0%), followed by *A. arenaria* vegetation (28.2%)
and a mixture of both plant species (21.8%) in August 2016. Species composition of the dunes changed
along a gradient from sea to land. Close to the sea dunes were vegetated by *E. juncea*, while, further
from the sea, dunes were also vegetated by *A. arenaria* alone, or in a mix with *E. juncea* (Fig. 3).
Landward of the foredune dunes were also vegetated by *E. juncea*, *A. arenaria* alone, or a mix of both
species. The foredune bisecting our study area was mainly vegetated with *A. arenaria*.
# Figure 3 approximately here #
In August 2016 dune area, volume and maximum height differed significantly between nebkha
dunes differing in species composition (volume: $F_{2,426}=3.02$, p=0.049; max. height: $F_{2,426}=58.8$, p <
0.001), but did not differ between dunes contrasting in shelter. Dunes with a mix of *E. juncea* and *A.*
*arenaria* had overall the highest volume and maximum height, whereas dunes with *E. juncea* had the
lowest volume and height. Dunes with *A. arenaria* had the largest range in dune volume (Fig. 4A, B,

C). For dunes with *E. juncea* seaward from the foredune the distance between nebkha dunes was higher, and thus clustering lower, than for to dunes with *A. arenaria* and dunes with both species ($F_{2,426}=51.5$, $p<0.001$). The dune volume did not significantly differ between dunes seaward and landward from the foredune (volume: $F_{1,426}=0.75$, $p=0.39$). In contrast, the dune height above NAP as well as the degree of clustering (Fig. 4D) were significantly higher for dunes landward from the foredune (dune height: $F_{1,426}=15.9$, $p<0.001$, clustering: $F_{1,426}=70.2$, $p<0.001$); we cannot exclude that part of these effects were related to the slightly older age (max. 5 years) of the nebkha dunes landward of the foredune.

# Figure 4 approximately here #

For the statistical model with relative change in dune volume as response variable, we had to correct for species distribution and spatial autocorrelation. We created a grid, with blocks of 50 m x 50 m, and we selected dunes that fell within a block. In total, we selected 236 dunes, which consisted of 41.95% of dunes with *E. juncea*, 36.02% of dunes with *A. arenaria*, and 22.03% of dunes with both species. This subset of dunes had an overall lower dunes size compared to all the nebkha dunes in the dune field, but had overall similar dune morphology and vegetation characteristics (Supplementary data S3).

Vegetation characteristics depended on the plant species dominating the dunes and on the degree of shelter. Nebkha dunes with *E. juncea* had significantly the lowest vegetation density, nebkha dunes with *A. arenaria* the highest and nebkha dunes which consisted of both species had an intermediate vegetation density (Fig. 4E, $F_{2,426}=48.91$, $p<0.001$). Similar to vegetation density, nebkha dunes with *E. juncea* also had the lowest maximum plant height, whereas nebkha dunes with *A. arenaria* and

consisting of both species had the highest maximum plant height (Fig. 4F, $F_{2,426}=42.38$, p<0.001).

Nebkha dunes landward from the foredune had significantly higher vegetation densities compared to seaward dunes ($F_{1,426}=45.49$, p<0.001), which is probably caused the calmer conditions landward from the foredune, which benefits plant growth or the slightly older age of these nebkha dunes. There was no significant difference in maximum plant height between nebkha dunes seaward and landward from the foredune ($F_{1,426}=0.41$, p=0.52). Nebkha dunes with *E. juncea* had the smallest vegetation area ($0.35\pm0.047m^2$), nebkha dunes with mixed vegetation the largest vegetation area ($10.90\pm3.05$ m$^2$) and nebkha dunes with *A. arenaria* have an intermediate vegetation area ($7.25\pm4.18$ m$^2$). The vegetation area on a nebkha dune is larger landward from the foredune ($9.61\pm3.96$ m$^2$), compared to seaward of the foredune ($2.04\pm0.41$ m$^2$). The vegetation area was correlated to dune volume (linear regression: $t_{430} = 25.29$, $p < 0.001$), however this relationship was stronger for nebkha dunes landward from the foredune, compared to nebkha dunes seaward from the foredune ($R^2 = 0.99$ vs. $R^2 = 0.69$).

## 3.2 Change in nebkha dune number and volume

The number of nebkha dunes within the measurement area changed over time, with nebkha dune numbers declining over winter and increasing during summer. The degree of dynamics depended on season, species and degree of sheltering.

### 3.2.1 Summer

Of the 434 nebkha dunes present in August 2016, 22.36% appeared over summer (April – August).
Most of these new dunes (65.93%) were *E. juncea* nebkha dunes, 31.87% were *A. arenaria* nebkha
dunes and only 2.20% were mixed dunes. Most (73.63%) new nebkha dunes developed seaward from
the foredune and were quite small in size with a volume of $2.72 \pm 0.29$ m$^3$ (mean $\pm$ SE). We assumed
that most of these dunes established over the growing season, as the orthomosaic showed a large
amount of wrack line material (plant material, woody debris, rope etc.) in their polygon in November
and April. However we cannot exclude that part of the large increase in the smaller *E. juncea* nebkha
dunes over summer is a result of their poor recognition in November and April.

351       Over summer, most nebkha dunes increased in dune volume, including the foredune which

increased over summer with 0.28% per week, reaching a volume of 64,444 m$^3$ in August. Only 4.16%
of the nebkha dunes showed a small decrease in the volume with a mean of $-0.041 \pm 0.014$ m$^3$/week.
Changes in dune volume were positively related to the initial dune volume (Fig. 5A, t-value$_{428}$= 57.11,
p<0.001) and were higher for nebkha dunes seaward of the foredune compared to nebkha dunes
landward of the foredune, resulting in a significant effect of shelter (t-value$_{428}$=2.72, p=0.0069). The
absolute changes in dune volume were also positively related to vegetation area, however this
relationship depended on the sheltering (vegetation area*sheltering by foredune: t-value$_{428}$ = 25.29, p >
0.001). Nebkha dune vegetation area explained more variation in the change in dune volume for dunes
landward of the foredune, compared to dunes seaward of the foredune ($R^2$= 0.98 vs. $R^2$ = 0.36).

362       # Figure 5 approximately here #

Compared to the absolute change in dune volume, the relative change in dune volume $(m^3/m^3/week)$ was mainly influenced by sheltering, with dunes seaward of the foredune growing faster than dunes to landward of the foredune (Fig. 6A). We found no significant difference in relative change in dune volume between dunes with different species composition (Fig. 6A, Table 1). In our statistical model plant height had a statistically significant effect on the relative dune growth. However, when tested in a single linear mixed model with block as random intercept, plant height had a $R^2$ of 0.0038, thus hardly explaining any variation in relative dune growth (Table 2). Several dune size variables were significant, but the individual variation explained by initial dune volume and dune height was very low, their $R^2$ ranging between 0.05 – 0.0033. The significant interactions between variables were mostly caused by the slight correlations between the explanatory variables. The clustering of nebkha dunes (i.e. the average height within 25 m of each dune) did not significantly affect the relative dune growth. We tested whether the effect of clustering was masked by the use of blocks as random intercept, since the amount of clustering was different between the blocks. We re-analysed the data without the blocks as random factor and again found no effect of clustering on the relative growth rate of dunes.

# Figure 6, Table 1 & 2 approximately here #

**3.2.2 Winter**

Over winter (November – April) 7.85% of the 344 nebkha dunes disappeared, of which 40.74% were dunes with *E. juncea*, 55.56% were dunes with *A. arenaria* and 3.70% were dunes with both species. These nebkha dunes disappeared both seaward (40.74%) and landward (59.26%) from the foredune and were overall quite small with an average volume of $2.23 \pm 0.19$ $m^3$.

Despite the decreasing number of nebkha dunes over winter, dunes increased in volume, the large foredune even increased with 0.22% per week. However on average the change in absolute dune volume was less positive than over summer, 21.30% of the dunes decreased -0.061±0.015 (SE) $m^3$/week in volume, particularly seaward of the foredune. 25.00% of these decreased dunes were covered with *A. arenaria* , 50.00% with *E. juncea* and 25.00% with both species. The absolute change in dune volume between November and April was positively related to the initial dune volume in November (Fig. 5B, t-value$_{428}$=2.12, p=0.034), but was only significant for dunes landward of the foredune. Dunes seaward of the foredune showed no relationship between absolute change in dune volume and the dune volume in November (shelter: t-value$_{428}$=-3.00, p=0.0029). Similar to initial dune volume, the vegetated area only explained variation in dune volume for the dunes landward from the foredune (vegetated area * sheltering by foredune: t-value$_{428}$ = 16.17, p<0.001).

The relative change in dune volume was influenced by species composition and degree of shelter (Table 1). Nebkha dunes with *E. juncea* increased relatively less in volume than dunes with *A. arenaria* (Fig. 6B); this effect was only significant for dunes seaward of the foredune.  We found no significant relationship between relative change in dune volume and vegetation density or maximum plant height (Table 2). There was a significant interaction between vegetation density and sheltering by the foredune, which could be related to the higher vegetation density at the dunes landward of the foredune. Initial dune volume, and sheltering, had significant negative effects on the relative change in dune volume, whereas clustering had a positive significant effect, but the relationships were very weak ($R^2$ between 0.002 – 0.05).

**3.3 Net nebkha dune growth**

Taken over the whole observation period November – August, the absolute nebkha dune growth ($m^3$/week) was higher at the seaward side of the foredune than at the sheltered landward side (slope seaward dunes: 0.37%, slope landward dunes: 0.25%, dune volume*position from foredune: t-value$_{428}$ = -11.7, p<0.001). Similarly, the relative dune growth ($m^3$/$m^3$)/week of the seaward dunes was also slightly higher than the landward dunes (seaward dunes: $0.27 \pm 0.00009$ (means±SE), landward dunes: 0.026±0.0001, F-value$_{1,230}$ = 18.51, p<0.001).

**3.4 Accuracy of photogrammetric reconstruction**

We checked the accuracy of the photogrammetric reconstruction by measuring the vertical error, the repeatability of the method and the degree in which NDVI predicted the biomass of the vegetation. The average vertical error was $7.3 \pm 0.2$ cm, with 80% of the measured points having a vertical error between -10 and 10 cm (Fig. S4.1). The vertical error increased with increasing distance from a ground control point. The vertical error increased up to 20 cm for points that were 150 m from a ground control point (Fig. S4.2). A vertical error of 10 cm could result in a deviation 3 – 6% in the dune volume, whereas the vertical error of 20 cm would result in a deviation of 5 – 12% in the dune volume (Table S4.1). The deviation depends however on the average elevation of a dune, a nebkha dune with a higher average elevation will have lower deviation of the vertical error than a nebkha dune with a low average elevation.

The source of error due to different conditions during consecutive mapping campaigns was
limited (Table S4.2). The difference between the DSMs of different flights with the same flight paths at
the same day was on average $3.9 \pm 3.9e^{-6}$ cm, with 80% of the raster cells of the DSM had a difference
between -0.07 and 0.07 cm (Fig. S4.3).
The degree in which NDVI represented vegetation biomass differed between species. The
summed NDVI of a nebka dune with *A. arenaria* showed a trend with the biomass of *A. arenaria* ($t_4 =$
2.43, p = 0.07, $R^2$ = 0.6), for nebkha dune consisting of *E. juncea* the summed NDVI was not
significantly related to the biomass of the vegetation ($t_5 = 1.43$, p = 0.21, $R^2 = 0.29$).
**4. Discussion**
The aim of this study was to explore the contributions of vegetation and dune size (i.e. initial dune
volume) to nebkha dune development expressed as change in dune volume. In addition, we were
interested in how the effects of vegetation and dune size on nebkha dune development were modified by
the degree of shelter. Our results show that the contribution of vegetation and dune size depended on
season and degree of shelter. In summer dune volume change ($m^3$/week) was explained by initial dune
volume and to a lesser extent by dune height, while species composition, vegetation height or density
had no effect. In winter dune volume change was explained by vegetation and initial dune volume,
depending on the degree of shelter. Exposed nebkha dunes with sparsely growing *E. juncea* grew less in
volume than exposed nebkha dunes with densely growing *A. arenaria*. In contrast, growth of sheltered
nebkha dunes was a function of initial dune volume. These findings are the first to show that the effect
of vegetation and dune size on the nebkha dune development depends on season. These results can be
used to improve modelling of coastal dune development.
**4.1 Dune size**
**4.1.1 Summer growth**
We found a positive linear relationship between the initial dune volume and the absolute change in dune
volume over summer. It is known that nebkha dunes affect sedimentation by changing the wind flow
patterns (Dong et al., 2004; Li et al., 2008). Previous studies have found that with increased dune
volume the area where the wind speed is reduced increases, which result in higher sedimentation rates
(Hesp, 1981; Hesp and Smyth, 2017). The linear relationship between initial dune volume and dune
volume change found for the nebkha dunes in our study indicates that different dune sizes have similar
effect on the wind flow pattern per unit of area, which indicates scale invariance (Hallet, 1990). Scale
invariance has been used for modelling nebkha and foredune development (Baas, 2002; Durán Vinent
and Moore, 2013), but not yet been validated for nebkha dunes to our knowledge.
Our study focussed on a relatively small range in nebkha dune sizes. It is likely that the linear
relationship between dune volume change and dune size will saturate when dunes continue to grow and
processes other than wind speed reduction become important. The latter is supported by the volume
change of the low foredune bisecting our study area. Over summer the large foredune increased 0.28%
per week in volume, which is much lower than the overall increase of 0.81% per week of the dune
seaward of foredune. Therefore, we expect that there is a critical dune size at which the relationship
between dune volume and absolute dune growth is no longer linear. However, what exactly the critical
dune size is, is difficult to predict, it probably depends on multiple factors such as available sediment
supply and vegetation growth. The wind flow patterns are not only influenced by dune volume, but also
by maximum dune height (Walker and Nickling, 2002). In our study we found a significant, albeit weak
effect of the maximum dune height on the relative growth, suggesting differences in height did not have
a large effect on the wind flow pattern and the subsequent deposition of sand.
The positive linear relationship between dune volume and dune growth was modified by
sheltering; dunes landward of the foredune increased 0.60% per week less in volume than dunes
seaward of the foredune. This reduction in dune growth rate is likely the result of decreased sand supply
landward of the foredune; presumably a large amount of the sand was captured by the foredune as was
also observed for other foredunes (Arens, 1996). In our study the decrease in sand transport was less
sharp as observed by Arens (1996), however the difference in foredune sink strength between the
foredune in our study and those measured in Arens (1996) could be related to its smaller size, its
relatively low height and/or its sparse vegetation cover of 29% (Keijsers et al., 2015). Clustering of
dunes did not have any significant effect on the relative growth rate, which suggests that these smaller
dunes do not significantly reduce the sand supply to the landward situated dunes.

**4.1.2 Winter**

In winter initial dune size was only a good predictor for growth of the nebkha dunes occurring landward
of the foredune. For these sheltered dunes, increases in volume (m$^3$/week) again followed a linear
relationship with initial dune volume. The absence of a relationship between initial dune volume and
dune growth for the exposed dunes occurring seaward form the foredune, suggests that dune erosion is

less dependent on initial dune size than dune growth. Dune erosion has mainly been attributed to wave run-up during storms (Haerens et al., 2012; Vellinga, 1982). Therefore, it seems reasonable to assume that the degree of erosion depends on whether the dune can be reached by high energy waves. Large dunes that are reached by high water levels can erode substantially, whereas small dunes can have no erosion if they are protected by other dunes from the high water.

Interestingly, the sheltered nebkha dunes had a slightly higher dune growth in winter compared to summer. This increase in dune growth for sheltered nebkha dunes can perhaps be explained by more frequent and/or intensive aeolian transport events during winter resulting into higher sand supply to the sheltered dunes (Davidson-Arnott and Law, 1990).

**4.2 Vegetation**

Vegetation characteristics were a poor predictor of dune volume change over the summer period, but were a significant predictor for dune volume change over winter. Over summer dune growth did not differ between nebkha dunes covered by different dune building plant species when corrected for dune size. Similarly, we did not find a clear effect of vegetation density and plant height on dune growth. This results contrast with other studies that report a significant difference in the ability of species to trap sand mediated by differences in shoot density and cover (Keijsers et al., 2015; Zarnetske et al., 2012). Perhaps the discrepancy with our study can be explained by the differences in spatial scale used between studies. We studied dune volume change at the scale of a nebkha dune including its shadow dune, whereas the other studies focussed on the scale of the vegetation patch (Bouma et al., 2007; Dong et al., 2008; Hesp, 1981, 1983; Keijsers et al., 2015; Zarnetske et al., 2012), where species specifics

effects are probably more pronounced than at the scale of the whole dune. Our results support findings

of Al-Awadhi and Al-Dousari (2013) who reported that the effects of vegetation on dune growth are

scale dependent for coastal nebkha dunes. They found that the linear relationship between shrub

vegetation characteristics and dune morphology levels off for bigger dunes. In our statistical models we

selected the smaller nebkha dunes, which was a consequence of only selecting dunes that were located

within one block. However even for these smaller nebkha dunes vegetation had no significant effect on

relative dune growth.  The vegetated area of the nebkha dunes did have a positive relationship with the

change in dune volume, however this relationship could be caused by collinearity between the vegetated

area and dune size, big dunes generally having a higher vegetated area. Since initial dune volume was

generally a better predictor for change in dune volume than the vegetated area, our results suggest initial

dune volume to be the better predictor for modelling.

Over winter nebkha dunes with *E. juncea* had a significantly lower relative growth rate than

nebkha dunes with *A. arenaria*, presumably because of their higher sensitivity to erosion. This species-

effect might be related to the sparser growth form of *E. juncea* in comparison to *A. arenaria* as dense

vegetation has been found to reduce the amount of dune erosion, by more effective wave attenuation

(Charbonneau et al., 2017; Koch et al., 2009; Silva et al., 2016). However, the effect of vegetation

density was not significant in our model suggesting that the species effect might be due to other species

differences, such as differences in rooting pattern. Another explanation is that the vegetation density

measurement did not reflect the real vegetation density, *E. juncea* was difficult to detect due to the low

NDVI values. The species effect was only significant for dunes situated at the exposed, seaward side of

the foredune where erosion by water likely occurred during the single storm covered by our study

period. Despite being statistically significant, the differences in relative growth rate between exposed nebkha dunes with *A. arenaria* and *E. juncea* was not very large. Nevertheless the species effect might become more pronounced with higher erosion pressure during more stormy winters (Charbonneau et al., 2017).

Interestingly, our species did show differences in dunes size. On average, nebkha dunes with *A. arenaria* were higher than nebkha dunes with *E. juncea*, that were broader (Bakker, 1976; Zarnetske et al., 2012). This difference in nebkha dune morphology suggests a higher sand catching efficiency of *A. arenaria*, as also suggested by (Zarnetske et al., 2012), this difference in sand catching efficiency might have been masked by including the initial dune volume and maximum dune height as explanatory variables. . We explored whether there is an effect of species composition on the change in maximum dune height over summer, but found no consistent effect. Perhaps the difference in nebkha dune morphology could be a result of differences in erosion between the nebkha dunes with different species composition over winter.

**4.3 Application of UAV monitoring for nebkha dune development**

Measurements on the accuracy of the photogrammetric reconstruction shows that the vertical error is between 0 cm – 20 cm, where most of the DTM pixels have a vertical error between 0 cm – 10 cm, resulting into a deviation of dune volume between 3 – 12%. We do not expect this variation to affect our results however, since the measurement error is random in nature and not systematic making explanatory variables less significant rather than more significant. The vertical error increased with

increasing distance from the ground control markers, for future studies a maximum distance of 70 m from each raster pixel to a ground control marker would be better than the 150 m we used. In our statistical models for relative dune volume change ($m^3/m^3$/week) we accounted for the increasing vertical error with increasing distance from the ground control marker by including blocks as a random factor, since the nebkha dunes within a block have similar distances to a ground control marker.

The vegetation density, expressed as NDVI/$cm^2$ dune, was not significantly correlated with the biomass. The poor relationship is likely a result of the low sample size (six or seven samples), in combination with the high contribution of non-green parts, such as stems and dead litter, that give no or weak NDVI signal. Since stems and dead litter do affect the wind flow pattern and attenuate waves, the poor relationship between NDVI and biomass could explain why we did not find an effect of vegetation density on dune growth and erosion. We did not measure the accuracy of the plant height, and can therefore not say how well the maximum plant height represents the real plant height, however it is probably an under-representation, since outliers are removed during photogrammetric processing.

**4.4 Implication for dune development**

**4.4.1 Net dune growth**

Exposed nebkha dunes had an overall higher net growth compared to sheltered nebkha dunes, indicating that summer growth offset winter erosion in our study period which was characterised by an average

summer and calm winter. This balance might have been different if winter conditions had been more severe.

During winter, storms determine the erosion of nebkha dunes seaward of the foredune. Multiple low intensity storms can lead to more erosion than one high intensity storm (Dissanayake et al., 2015; Ferreira, 2006; van Puijenbroek et al., 2017). Whether exposed dunes have a higher net dune growth compared to dunes landward from the foredune depends mainly on the storm intensity and frequency. A single high intensity storm can erode all the sand that exposed dunes have accumulated over a whole summer, and in such case sheltered dunes could have a higher growth rate than the exposed dunes. The exact relative growth rate over summer depends on the number of aeolian transport events. Linking the number of aeolian transport event to the relative growth rate over summer would be a worthwhile avenue for future research.

Sand supply and storm intensity are also affected by local conditions as beach morphology. A minimum beach width is needed to reach maximum aeolian transport, the fetch length (Delgado-Fernandez, 2010; Dong et al., 2004; Shao and Raupach, 1992). Our study site had a wide beach (0.9 km wide), and we assume that the maximum aeolian transport was reached. The net growth of our foredune was approximately 30 m$^3$ per m foredune parallel to the sea for a period of 10 months. This growth rate does also occur at other places along the Dutch coast, but is not very common (Keijsers et al., 2014). Storm intensity is also influenced by beach morphology. The presence of intertidal bars and a wide beach can reduce the storm intensity by wave attenuation (Anthony, 2013; Ruggiero et al., 2004). Therefore, we can assume that the net dune growth we found in our study will depend on the beach

morphology. On smaller beaches we expect the net dune growth to be lower compared to wider
beaches, due to the lower sand supply by reduced fetch length and higher storm erosion of dune (van
Puijenbroek et al., 2017)
**4.4.2 Vegetation**
For coastal dune development vegetation is essential, however the species-composition of the
vegetation seems less important than we assumed: species did not seem to affect dune growth over the
summer, but did affect dune growth over winter.

587          We did find differences in nebkha dune morphology between the species, which suggest a causal

relationship. However, the difference in nebkha dune morphology between species is probably also
caused by differences in nebkha dune age. In Western Europe, the primary succession of coastal dunes
is generally assumed to start with *E. juncea*. Only after a fresh water lens has developed in the dune
with *E.* juncea, *A. arenaria* will establish (Westhoff et al., 1970). Over time *A. arenaria* will
outcompete *E. juncea*. This assumed succession pathway matches part of the spatial patterns that we
found in our study site and explains why nebkha dunes with only *E. juncea* are relatively small. Over
time these small nebkha dunes merge together after which *A. arenaria* is assumed to establish.
However, we found that *A. arenaria* has a large range in dune volume suggesting that, contrary to
current assumptions, *A. arenaria* can also establish on the bare beach without *E. juncea*, as long as the
soil salinity is not too high.

598          At our study site only two dune building species occur, however there are many different dune-

building species. It could very well be that other dune building species do have significant differences in

the nebkha dune growth over summer. For further research it would be interesting to study if these

results are similar in another nebkha dune system with different plant species.

### 4.4.4 Application

To our knowledge, we are the first to report on the relationship between initial dune volume and dune

growth for nebkha dunes in the field. The linear relationship that we found in our studies can be

incorporated in mathematical models that predict dune development. Furthermore, our research shows

that for predicting dune growth species identity does not matter during the summer, however it does

matter during the winter. This indicates that for dune building models, species identity is especially

important when winter survival of nebkha dunes is modelled. Furthermore, for the construction of an

artificial dune it appears to be crucial to plant the more storm resistant species.

Despite the presence of smaller nebkha dunes seaward of the foredune, the foredune showed a

large increase in volume compared to similar foredunes along the Dutch coast. This indicates that sand

supply to the foredune was not seriously hampered by the presence of the small vegetated dunes, while

the smaller dunes seaward of the foredune likely added to the protection of the foredune against storm

erosion. For coastal management it could be beneficial for foredune growth to have nebkha dunes

seaward of the foredune given a high sand supply.

### 5. Conclusions

The purpose of this study was to explore the contribution of vegetation and dune size on nebkha dune development at locations differing in shelter from the sea. Our results show that 1) the contribution of vegetation and dune size depend on season and degree of shelter. 2) Species composition does not affect dune growth over summer, but does affect dune growth during winter, particularly at exposed sites. 3) During early dune development, nebkha dune growth is linearly related to nebkha dune volume, whereas dune volume does not seem to matter for nebkha dune erosion. 4) Sheltering by a foredune reduces both sand supply and dune erosion; the net effect of shelter on dune growth therefore likely depends on beach morphology and weather conditions. These results can be incorporated in models predicting nebkha dune development and can be used by managers to determine coastal safety.

**Acknowledgements** We would like to thank Ministry of Defence and Staatsbosbeheer to allow UAV flights in their nature area. We would like to thank the technology foundation STW (grant number STW 12689 S4) for funding the NatureCoast project, which made this research possible. Finally, the authors thank the reviewers Anne-Lise Montreuil and Patrick A. Hesp for their useful and extensive comments on a previous draft of the manuscript.

**Competing interests** The authors declare that they have no conflict of interest.

**Data availability** Final dataset used for the statistical tests and the data on the accuracy of the photogrammetric reconstruction are archived in the 4TU Datacentre http://researchdata.4tu.nl/home/, under https://doi.org/10.4121/uuid:8a2-30db-4328-bf04-40618bf31e4c. The RAW images, digital

surface model, digital terrain model and the orthomosaic are available upon request by the
corresponding author.
**Author contributions** MvP, CN and JS performed UAV flights and image calibration. MvP, CN and
JL analysed the data. MvP, CN, JS, AdG, MR, FB and JL provided guidance on the scope and design of
the project, and contributed to the writing of the manuscript.
**Supporting information**
Additional supporting information can be found in the online version of this article:
**Supplement S1** Weather conditions in our study site for 2013 - 2016
**Supplement S2** DTM, DSM and orthomosaic of each mapping campaign
**Supplement S3** Nebkha dune morphology of selected dunes
**Supplement S4** Accuracy photogrammetric reconstruction

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

**Table 1.** Statistical models for the relative change in dune volume between April – August (summer) and November – April (winter) for nebkha dunes. In this model we tested the effect of species, dune size, and degree of sheltering. The data was analysed with a general linear mixed model with blocks as random intercept. The standardized estimates and level of significance are shown for the models. Model selection was performed with AIC (Akaike information criterion) as selection criteria. Marginal $R^2$ is the variation explained by the fixed factors, whereas the conditional $R^2$ is the variation explained by the fixed and random factors.

| Model with species | Dependent variable: | | | |
|---|---|---|---|---|
| | Relative change in dune volume | | | |
| | Summer | | Winter | |
| | Full model | Model selection | Full model | Model selection |
| **Main effects** | | | | |
| Intercept | 1.18*** | 1.17*** | 0.92*** | 0.94*** |
| E. juncea | -0.02 | | 0.005 | -0.02** |
| Mix | 0.02 | | 0.02 | -0.003 |
| Dune volume | 6.10 | 8.27*** | -6.0* | -3.43** |
| Clustering | -0.22 | -0.18 | 0.22 | 0.23 |
| Max. dune height | -0.25 | -0.31* | 0.15 | 0.087 |
| Sheltering by foredunes | 0.29* | 0.31** | -0.31** | -0.31** |
| Interaction effects | | | | |
| E. juncea * Dune volume | 0.90 | | 1.90 | |
| Mix * Dune volume | -0.11 | | 1.41 | |
| E. juncea * clustering | 0.11 | | 0.04 | |
| Mix * clustering | 0.01 | | -0.006 | |
| E. juncea * max. dune height | -0.08 | | -0.09 | |
| Mix * max. dune height | -0.02 | | -0.033 | |
| E. juncea * Shel. by foredune | -0.05 | | 0.03 | |
| Mix * Shel. by foredune | -0.02 | | 0.001 | |
| Dune volume * clustering | -4.64* | -5.65** | 4.44** | 4.10** |
| Dune volume * max. dune height | -1.16 | -2.01* | 0.62 | |
| Dune volume * Shel. by foredune | 1.85 | 2.00* | -1.11 | -1.31* |
| Clustering * max. dune height | 0.31 | 0.34* | -0.29 | -0.27* |
| Clustering * Shel. by foredune | -0.12 | -0.17* | 0.12 | 0.13 |
| Max. dune height * Shel. by foredune | -0.20* | -0.18* | 0.19** | 0.19** |
| Marginal R² | 0.31 | 0.31 | 0.25 | 0.23 |
| Conditional R² | 0.34 | 0.33 | 0.39 | 0.39 |
| Observations | 236 | 236 | 236 | 236 |

| | -632.60 | -685.45 | -673.10 | -709.11 |
| Akaike Inf. Crit. | -632.60 | -685.45 | -673.10 | -709.11 |
| Bayesian Inf. Crit. | -555.08 | -641.04 | -595.57 | -661.35 |

Note:

*p<0.05; **p<0.01; ***p<0.001

802

803

**Table 2.** Statistical models for the relative change in dune volume between April – August (summer) and November – April (winter) for nebkha dunes. In this model we tested the effect of vegetation characteristics, dune size and degree of sheltering. The data was analysed with a general linear mixed model with blocks as random intercept. The standardized estimates and significance values are shown for the models. Model selection was performed with AIC as selection criteria. Marginal $R^2$ is the variation explained by the fixed factors, whereas the conditional $R^2$ is the variation explained by the fixed and random factors.

| *Model with vegetation characteristics* | *Dependent variable:* | | | |
|---|---|---|---|---|
| | Relative change in dune volume | | | |
| | *Summer* | | *Winter* | |
| | Full model | Model selection | Full model | Model selection |
| **Main effects** | | | | |
| Intercept | 1.24*** | 1.24*** | 0.90*** | 0.81*** |
| Vegetation density | -0.003 | | -0.05 | -0.03 |
| Max. plant height | 0.15 | 0.14** | 0.04 | |
| Dune volume | 8.65*** | 6.62*** | -2.72 | -3.67** |
| Clustering | -0.21 | -0.23 | 0.29 | 0.40** |
| Max. dune height | -0.44* | -0.41** | 0.07 | 0.17 |
| Sheltering by foredune | 0.26* | 0.29* | -0.28* | -0.25** |
| | | | | |
| Veg. density * max. plant height | -0.01 | | 0.001 | |
| Veg. density * dune volume | 0.83 | | 0.92 | |
| Veg. density * clustering | -0.03 | | 0.078 | 0.06 |
| Veg. density * max. dune height | 0.04 | | -0.03 | |
| Veg. density * Shel. by foredune | -0.005 | | -0.03 | -0.04** |
| Max. plant height * dune volume | -0.58 | | -0.19 | |
| Max. plant height * Clustering | 0.02 | | -0.06 | |
| Max. plant height * max. dune height | -0.11 | -0.10** | 0.04 | |
| Max. plant height * Shel. by foredune | 0.004 | | -0.01 | |
| Dune volume * clustering | -6.37** | -6.30*** | 4.51** | 4.65*** |
| Dune volume * max. dune height | -1.54 | | -1.11 | |
| Dune volume * Shel. by foredune | 1.63 | 1.95* | -2.23* | -1.82** |
| Clustering * max. dune height | 0.40* | 0.41** | -0.32 | -0.42** |

| | | | | |
|---|---|---|---|---|
| Clustering * Shel. by foredune | -0.15 | -0.17* | 0.05 | |
| Max. dune height * Shel. by foredune | -0.16 | -0.16* | 0.28** | 0.31*** |
| Marginal R² | 0.33 | 0.31 | 0.24 | 0.21 |
| Conditional R² | 0.37 | 0.35 | 0.42 | 0.40 |
| Observations | 236 | 236 | 236 | 236 |
| Akaike Inf. Crit. | -622.85 | -674.05 | -656.46 | -704.97 |
| Bayesian Inf. Crit. | -542.07 | -626.28 | -575.68 | -657.20 |

*Note:*
[*]p<0.05; [**]p<0.01; [***]p<0.001

811

812

**Figure captions**

**Fig. 1** A) Overview of the Hors on Texel, the Netherlands. The white lines show the flight path for the four different flights. The points show the position of the ground control markers. The white polygon is the monitoring area, which is 200 m x 400 m. B) Photograph of the study site with the UAV used to monitor the nebkha dunes.

**Fig. 2** Workflow of the methodology. The 3D point cloud from the photogrammetry was used to construct a DSM, DTM and NDVI orthomosaic. The DTM and NDVI orthomosaic where used to define the nebkha dunes. The explanatory variables for the statistical models were derived from the DSM, DTM and NDVI orthomosaic. For a more detailed explanation see methods.

**Fig. 3** Overview of the monitoring area. A) The elevation is shown with the Digital Terrain Model (m NAP), the green pixel indicates grass cover and the polygons indicate the nebkha dunes. B) The colour indicates the species present on the nebkha dune and the squares the blocks. The foredune in the middle of the monitoring area is excluded from the statistical analysis. Some dunes were cut-off by the edge of the DTM, we discarded these dunes.

**Fig. 4** Different dune characteristics for nebkha dunes in August with *A. arenaria*, *E. juncea* and a mix of both species separated for dunes seaward and landward of the foredune: A) Dune area, B) Maximum dune height, C) Dune volume, D) Clustering: mean height around a 25m radius around the dune, E) Vegetation density, F) Plant height. The letters denote the significant difference between the bars. Seaward of the foredune there were 41 dunes with *A. arenaria*, 193 dunes with *E. juncea*, and 53 dunes with both species, landward of the foredune there were 81 dunes with *A. arenaria*, 23 dunes with *E. juncea*, and 41 dunes with both species. NAP refers to Amsterdam Ordnance Date, which refers to mean sea level near Amsterdam

**Fig. 5** The relationship between initial dune volume ($m^3$) and the absolute change in dune volume ($m^3$/week) for: A) summer (April – August); B) winter (November – April). The data is shown for nebkha dunes seaward and landward of the foredune. The black line shows the regression prediction, the grey dashed line the 95% confidence interval. The formulas are the result of a linear regression model.

**Fig. 6** Relative change in dune volume ($m^3/m^3$)/week for nebkha dunes with *A. arenaria*, *E. juncea* and a mix of both species and separated for dunes seaward and landward of the foredune for: A) summer, April – August; B) winter, November – April. The letters denote the significant difference between the bars. Seaward of the foredune there were 28 dunes with *A. arenaria*, 77 dunes with *E. juncea*, and 28 dunes with both species, landward from the foredune there were 57 dunes with *A. arenaria*, 22 dunes with *E. juncea*, and 25 dunes with both species.

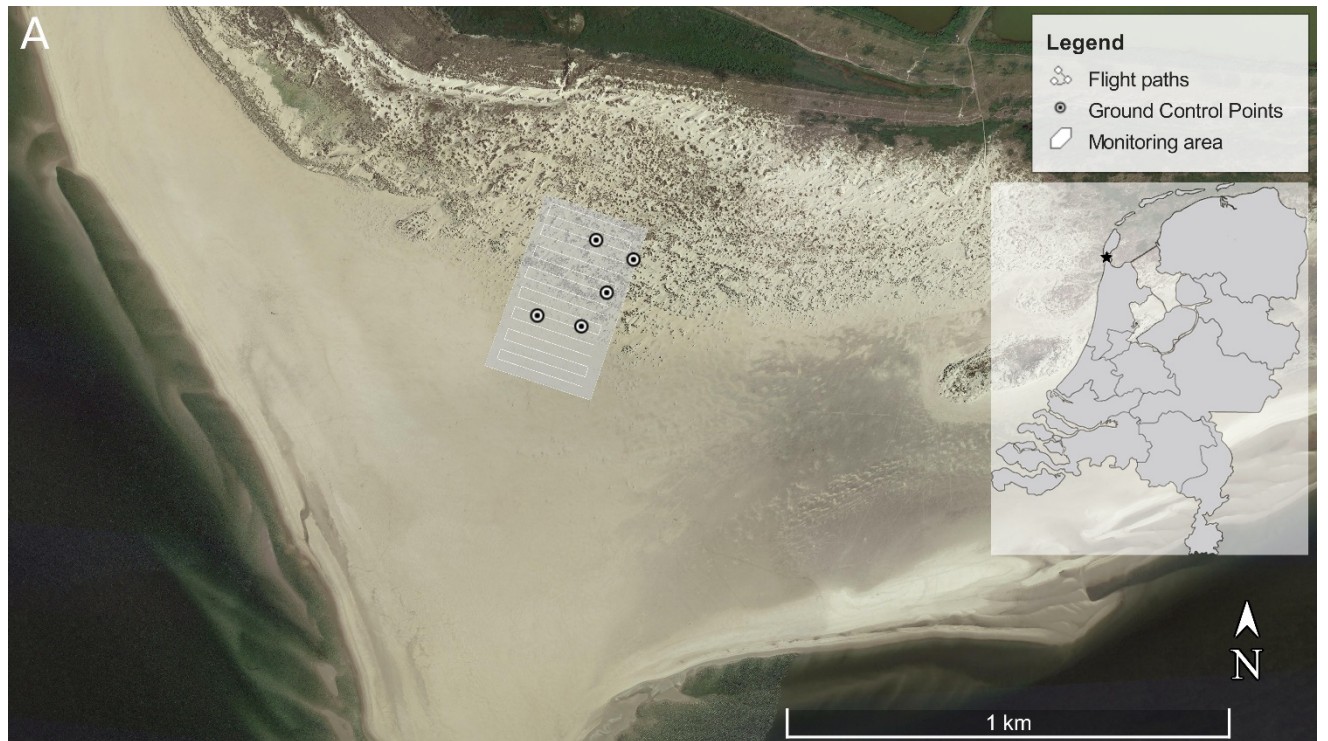

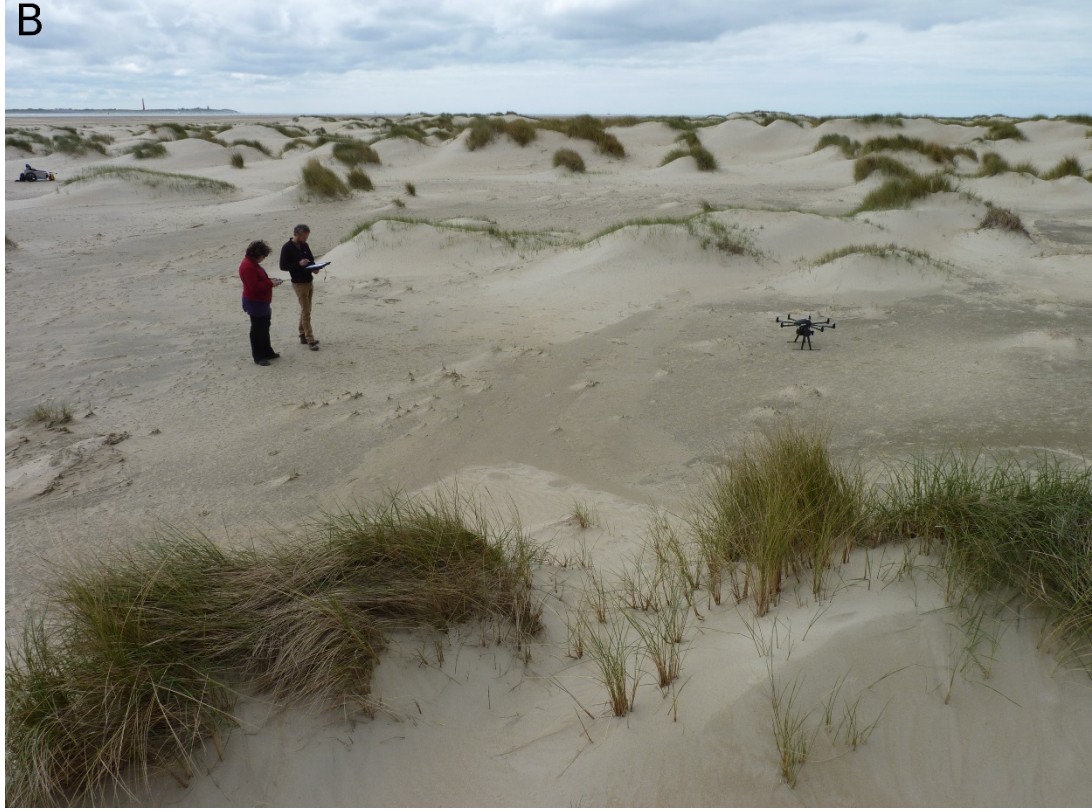

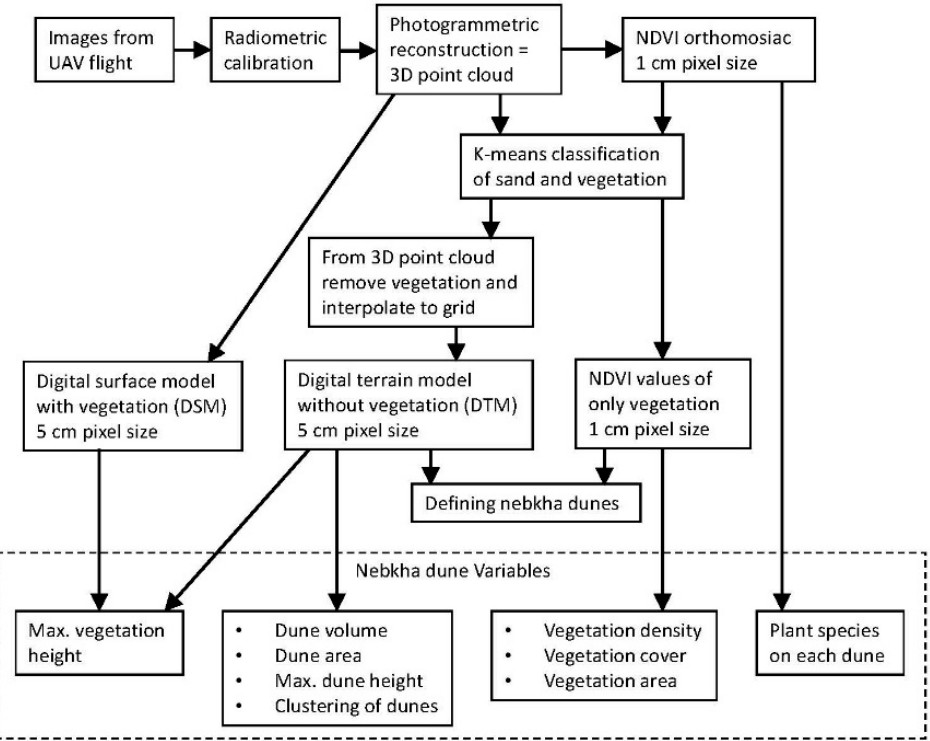

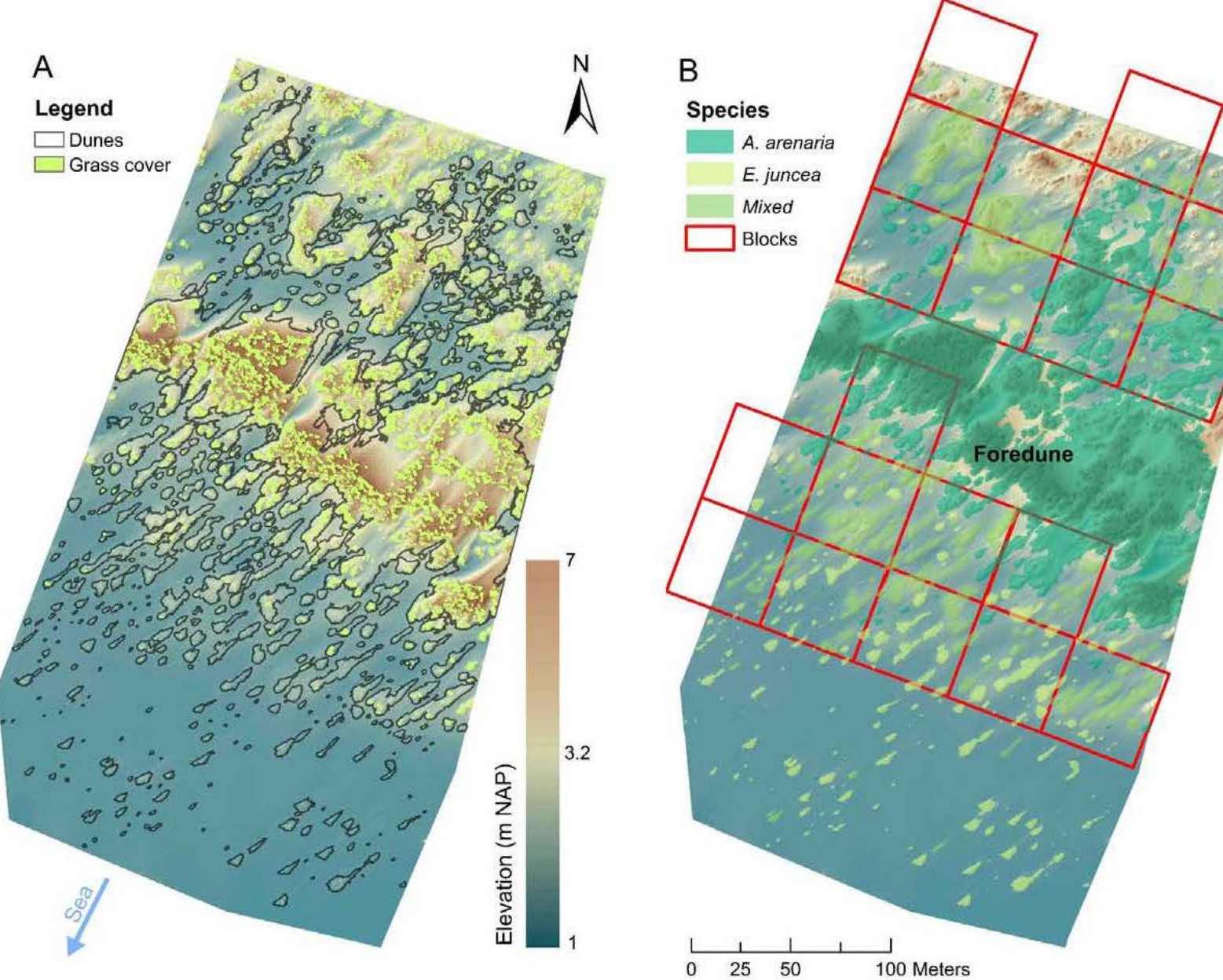

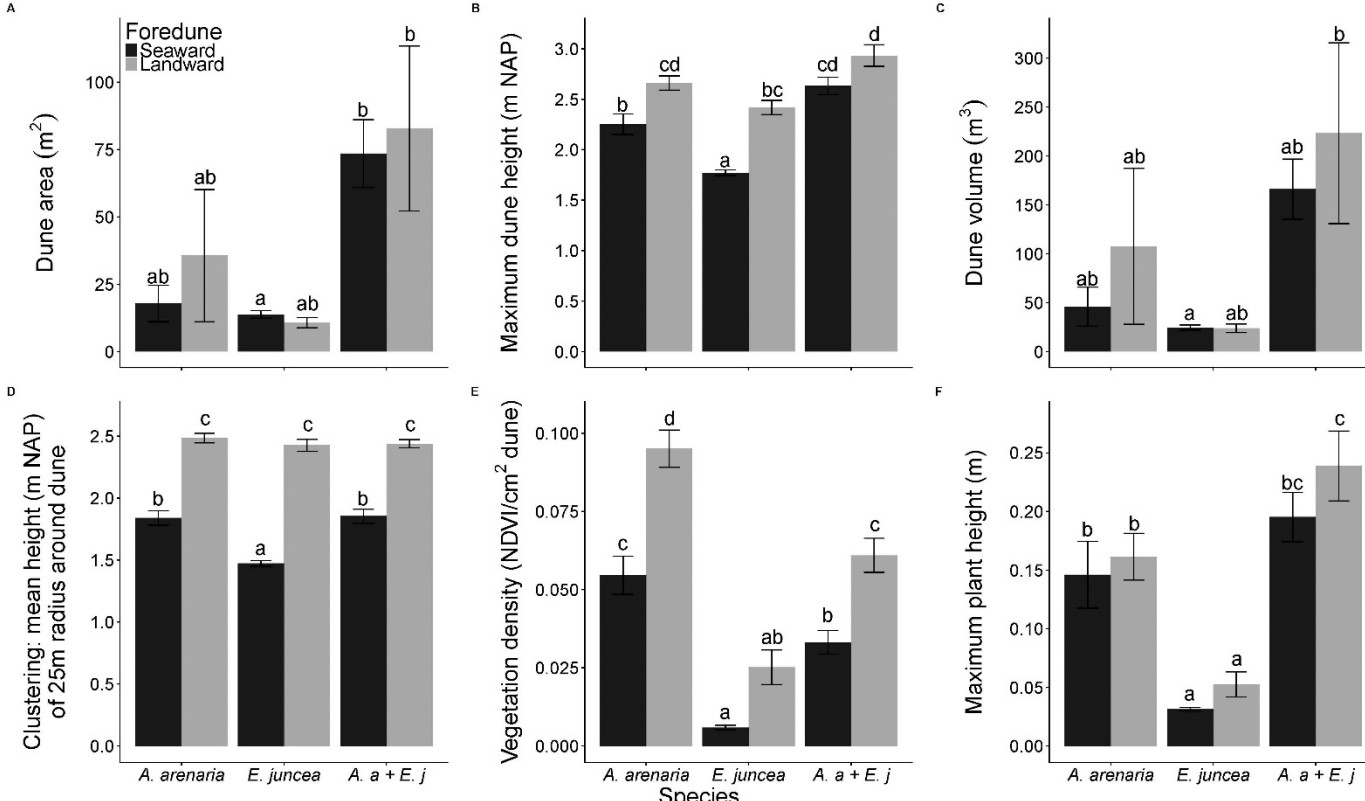

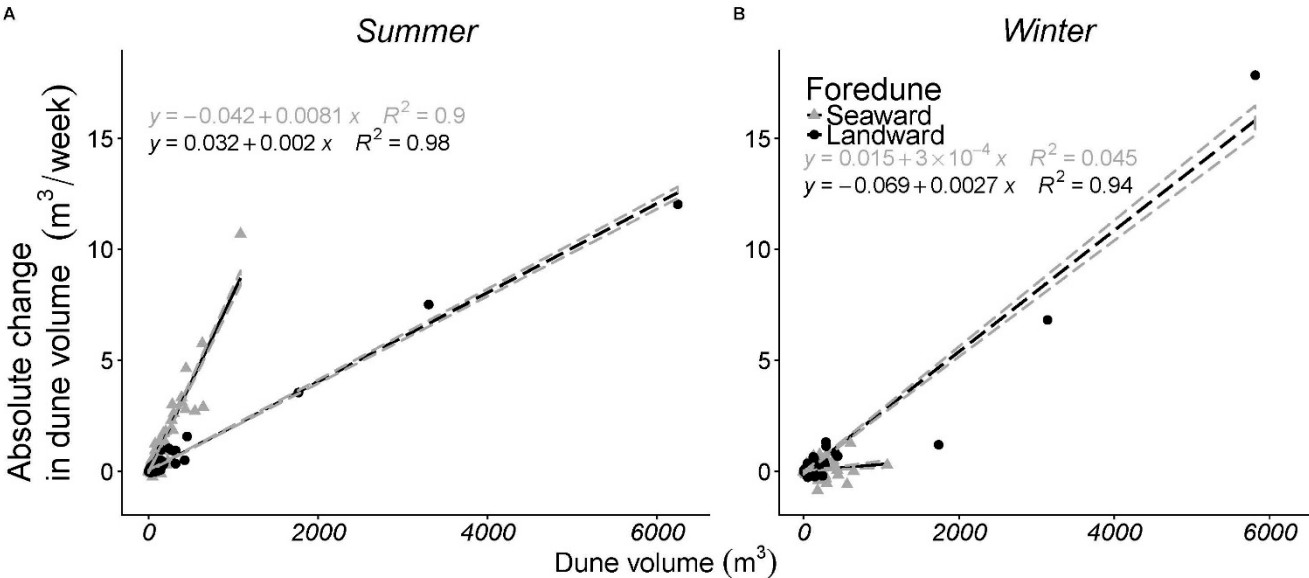

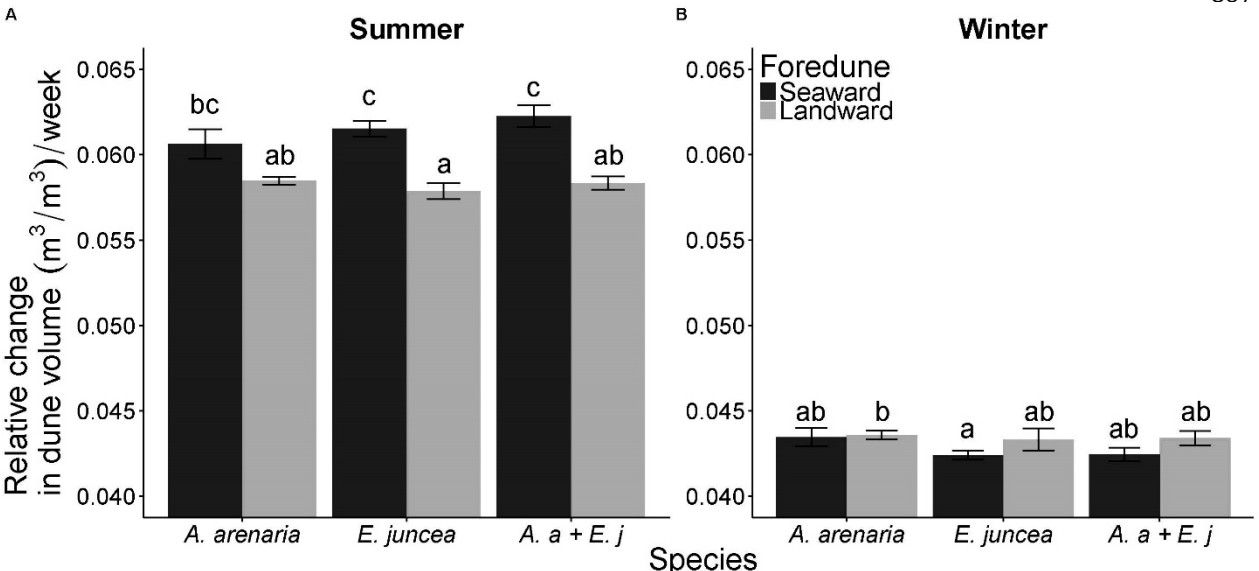