# Peer review of "Exploring the contributions of vegetation and dune size to early dune"

_Biogeosciences, 2017_

## Referee Comment (RC1) · A.-L. Montreuil (Referee) · 13 Jun 2017

This is a very interesting piece of work that assesses the relationships between vegetation and dune morphology based on UAV surveys. The authors successfully follow the contributions of vegetation and dune morphology to dune development on a large beach in the Netherlands. A truly interesting part of this study is the fact the dune growth is determined in summer and winter by dune size and vegetation respectively. I believe the paper is a valuable contribution, and I think it should be published after the authors have clarified/reviewed a few points. I have no issues with the work per se

in terms of the statistical analyses applied to relate dune volume with both vegetation species and characteristics. However, I have some moderate comments regarding the analysis of the UAV acquisition and processing.

Moderate concerns:

There are only 5 ground control points used which are not homogenously located in the investigated study site (e.g. not in each corner and middle of the site). Thus, my concern is that the sum of error from data acquisition to DTM generation is likely to be above 5cm. Also the error of the DSM for each survey is likely to be different due to difference of weather conditions and survey acquisition. I would suggest the authors to report the error of the DSM of each survey.

Unfortunately no field vegetation height surveys are reported to be carried out during the UAV flight. Could the authors report the error of vegetation height extracted by the difference between DSM and DTM? I would expect a difference between summer and winter since the vegetation binding is likely to be higher for the latter.

I would suggest the authors to be critical about the limitations of their technique.

I think that it would be nice if the authors present the DTM, DEM and orthomosaics for each survey in a Figure. This could help further to support the analysis.

Minor comments:

- In the abstract, some result values should be added to support the interpretation of the findings. I would not suggest to have biogeomorphology as a keyword because it is not mentioned in the text of the manuscript.

- I would suggest to modify Figure 1 by: adding a ground picture where dunes, and vegetation could be visualized and locating the foredune.

-The methodology section is quite long. I would suggest to have a separate study site section. Also I think that it would be easier for the reader to have a figure of the

workflow of the methodology.

- Could the authors justify the thresholds used to define the dunes in lines 184-185.

- Authors said that there are 11 blocks landward from the foredune in line 236 . However only 10 blocks could be seen in Figure 2.

- In Figure 4, the markers for seaward and landward cannot be differentiated. They should not be the same.

I truly enjoyed the discussion part. I checked the references and found them all correct and found them all correct (i.e. references cited in the text are in the list and vice versa). The manuscript is well written. I could not find grammatical errors or awkward sentences that would distract me as a reader. On the contrary, the text is easy to follow. I believe the manuscript should proceed to publication after the revision outlined above. UAV systems are becoming more and more accessible to a wider community and hence I believe contributions such as the one outlined in this paper will be welcome by a number of other coastal researchers.

All the best

Anne-Lise Montreuil

---

## Referee Comment (RC2) · P. A. Hesp (Referee) · 10 Jul 2017

This is an interesting topic , sadly very poorly written.

Line 55- there are multiple papers outlining how incipient or embryo dunes develop in multiple countries so this is patently wrong – remove or rephrase.

Lines 57 to 63- actually Hesp stated that incipient foredunes are initiated in several ways and by nebkha and shadow dune formation is only ONE way. If the authors are going to review how incipoinet foredunes are formed they need to state all the other

ways too – e.g. by aeolian deposition in continuous alongshore canopies of vegetation as well as discrete nebkha. And its: incipient foredunes" NOT incipient dunes" - the latter describes any type of dune...

Lines 79-80 these refs are very recent – the more comprehensive reviews of e.g. effect of veg density and distribution are in hesp papers – 1983, 1988 for example so cite these and Arens papers.

Lines 91-92. You need to explain better WHY u think greater dune size should mean greater accretion/deposition. Is it because u think if a dune is big then it obviously has a greater sediment supply than a small dune? BUT what about age? How has this been taken into account? A dune might be small because its young/in early development stage, a big one because its been sitting there for 200 years or gets regular scarping, scarp fill, crest growth due to that... Also is it because a larger vegetation patch would produce a larger nebkha and therefore would be able to collect more sand? There are multiple answers here and you must discuss there and later in the discussion/conclusions the impacts of these on your results.

Lines 92-93: WHY? Because of snow cover, more wave energy and erosion, wet sand WHAT? Please explain.

Lines 101-102: WHAT 3 types of dunes? You haven't said before this that there are 3 types. In line 100 u say dunes are formed by 1, 2 or a mixture... is that what u mean by saying 3 TYPES of dunes? In which case they are NOT types.(im convinced even by this stage you do not understand how dunes are classified...) they are ALL incipient foredunes formed in diff species or mixtures of species. REWRITE. Elucidate please!

It is NOT obvious until one gets into the methods section that you are mostly, or entirely talking about incipient foredunes and mostly nebkha and shadow dunes. You need to state this clearly at the start of the paper and also in the abstract.

Lines 275-276: dune ht - WHY? Because these are older since they are more landward? Explain 289-290: obviously because they formed earlier and are older and have had a greater time to collect sand. How about stating these kinds of associations when u state your results?

Also u are omitting the important papers on flow and sedimentation in patches or vegetation – classic study of diff patch density by Qian et al; Liu papers, Bouma paper on flow in veg patches underwater etc – these all provide excellent explanations of how density controls nebkha development and need to be reviewed and cited.

Bouma, T.J., van Duren, L.A., Temmerman, S., Claverie, T., Blanco-Garcia, A., Ysebaert, T., Herman, P.M.J., 2007. Spatial flow and sedimentation patterns within patches of epibenthic structures: Combining field, flume and modelling experiments. Continental Shelf Research 27, 1020–1045.

Dong.,Z., Wanyin, L., Guangqiang, Q., Ping, L., 2008.Wind tunnel simulations of the three-dimensional airflow patterns around shrubs. Journal of Geophysical Research 113: F202016, doi: 10.1029/2007JF000880

Lines 337-340: its strange and weird that you state dune vol is related to dune volume! Of course its is as it's the same thing. . . . Rewrite to explain better what you are correlating here.

Line 358: YOU MEAN: "The aim of this study was to explore the contributions of vegetation and dune size to NEBKHA dune development" - add this word otherwise its totally confusing and non-obvious what u are talking about; i.e. ANY dune development??!

Line 359 – now your aim is ONLY about degree of shelter? What about the other aims stated at the start of the paper??

Lines 368-369: because you have failed to adequately review the literature you are stating untruths here. One of the great papers to fully show how seasons control foredune growth is the one by Davidson-Arnott (ref in Hesp 2002 paper maybe) . Check his book which has the model in it I think. At any case remove the statement that this

is the first to relate foredune growth to seasonal change.

Lines 390-393: the referencing of the transverse dune lit here doesn't compute. Shadow dunes and/or nebkha do not at all have the same flow dynamics as transverse dunes. You need to rethink this entire idea and writing. Shadow dunes for example are controlled by paired horizontal flow vortices and max slope angle (hesp 1981). Nebkha vol and height is largely controlled by veg density and nebkha age and rate of plant growth. . ..

Lines 415-416- and less storm surge, wet high tide beach, etc on the sheltered side??

Line 418 and subsequent lines: You are NOT describing "veg characteristics" here. BE specific – u are at least first describing the effect of veg species differences or combinations of species, NOT density, distribution, height etc. So be specific – rewrite.

OK I see that you discuss these other factors next BUT would be better to still rewrite the first part to make it clear you are first just talking about species differences.Lines 448-449: there are several studies showing that ammophila does trap more sand generally compared to other species due to its high density clump-like nature so cite some of these.

lines 514-515: I don't see anywhere a decent explanation of why this is the case. You need to better explain this conclusion.

Cheers

Patrick Hesp

---

## Author Comment (AC1) · 2 Oct 2017

Thank you for your comments and the helpful feedback, which will help us improve both clarity and impact of the MS. Below we provide a point-by-point response to the comments, including their consequences for the MS.

Reviewer comments are indicated with open bullet points, whereas our response is indicated with a dash.

Kind regards, also on behalf of all co-authors

[Figure]

Marinka van Puijenbroek

Summary: This is a very interesting piece of work that assesses the relationships between vegetation and dune morphology based on UAV surveys. The authors successfully follow the contributions of vegetation and dune morphology to dune development on a large beach in the Netherlands. A truly interesting part of this study is the fact the dune growth is determined in summer and winter by dune size and vegetation respectively. I believe the paper is a valuable contribution, and I think it should be published after the authors have clarified/reviewed a few points. I have no issues with the work per sein terms of the statistical analyses applied to relate dune volume with both vegetation species and characteristics. However, I have some moderate comments regarding the analysis of the UAV acquisition and processing.

Moderate comments

o There are only 5 ground control points used which are not homogenously located in the investigated study site (e.g. not in each corner and middle of the site). Thus, my concern is that the sum of error from data acquisition to DTM generation is likely to be above 5cm. Also the error of the DSM for each survey is likely to be different due to difference of weather conditions and survey acquisition. I would suggest the authors to report the error of the DSM of each survey.

- Thank you for calling attention to this aspect. We set out to calculate three potential sources of error: 1) the vertical error associated with the use of photogrammetry, 2) the error involved in performing multiple campaigns and 3) the relationship between NDVI and vegetation biomass. Concerning 1) The vertical error of the DTM ranged between $0 - 20$ cm. This value did depend on the distance to the ground control marker, further from the marker the higher the vertical error. This vertical error means for the dune volume that there will be an error in dune volume between $5 - 12$ % depending on the vertical error. Concerning 2), the repeatability of the photogrammetric reconstruction was on average 3 cm. We do not expect the vertical error error to affect our results

however, since the measurement error is random in nature and not systematic making explanatory variables less significant rather than more significant. The third point is discussed in the comments below.

- We will add information on the accuracy of the photogrammetry reconstruction to the results and discuss the implications of the accuracy of the photogrammetric reconstruction for our results in our discussion.

- We will include in the supplementary data 1) graphs of the frequency distribution of the vertical error, 2) the relationship between distance of the ground control marker and the vertical error, and 3) a graph on the repeatability of the photogrammetric reconstruction. In the supplementary data we also included two tables of the deviation of the dune volume for different vertical errors and information about the dense point cloud for the different mapping campaigns.

o Unfortunately no field vegetation height surveys are reported to be carried out during the UAV flight. Could the authors report the error of vegetation height extracted by the difference between DSM and DTM? I would expect a difference between summer and winter since the vegetation binding is likely to be higher for the latter.

- Unfortunately we did not measure the vegetation height during our mapping campaigns and can therefore not report the error of the vegetation height. The maximum vegetation height calculated in our study is most likely an under-estimation, because the during photogrammetric reconstructions outliers are removed. During winter the maximum vegetation height will most probably be lower, partly because in the field the vegetation height is lower. The NDVI signal is also lower and this will also result in a lower maximum vegetation height, especially for the dunes covered with Elytrigia juncea, since their NDVI signal is very weak in winter.

- We did relate the summed NDVI per dune with the biomass of the vegetation per dune (see response earlier comment). We did not find a significant relationship between the NDVI and the biomass on a dune, but this was partly due to the low sample size.

Biomass also includes vegetation parts such as stems and litter, and these parts do not contribute (much) to the NDVI signal, which could explain the absence of a correlation. We will add this result to our manuscript.

o I would suggest the authors to be critical about the limitations of their technique.

- We agree that is it important to be critical about the limitation of the UAV monitoring, and therefore we will add a paragraph to the discussion, which discusses how the accuracy of the DTM could affect our results.

o I think that it would be nice if the authors present the DTM, DEM and orthomosaics for each survey in a Figure. This could help further to support the analysis.

- We agree, and will include a graph with the DTM, DSM and orthomosaic for each mapping campaign in a figure in the supplementary material.

Minor comments:

o In the abstract, some result values should be added to support the interpretation of the findings. I would not suggest to have biogeomorphology as a keyword because it is not mentioned in the text of the manuscript.

- We will add some result values in the abstract and remove biogeomorphology as a keyword.

o I would suggest to modify Figure 1 by: adding a ground picture where dunes, and vegetation could be visualized and locating the foredune.

- We will add a ground picture of the area, unfortunately there is no photograph from which we can clearly indicate the foredune.

o The methodology section is quite long. I would suggest to have a separate study site section. Also I think that it would be easier for the reader to have a figure of the workflow of the methodology.

- We will add a title above the study site section. We will add a figure with the workflow of our methodology.

o Could the authors justify the thresholds used to define the dunes in lines 184-185.

- We will add this sentence to justify the thresholds used to define dunes: The 5 cm threshold is the minimum that can be accurately derived from the images and corresponds with visual estimates of nebkha dune foot; Pixels above 5 cm indicated sand deposition, and a slope of 15° has been earlier identified by Baas et al (2002), as the slope for a shadow dune.

o Authors said that there are 11 blocks landward from the foredune in line 236 . However only 10 blocks could be seen in Figure 2.

- There are 11 blocks landward from the foredune, however in our figure 2 one block was cut off by the edge of the figure. We will change figure 2 to show all the blocks.

o In Figure 4, the markers for seaward and landward cannot be differentiated. They should not be the same.

- We will change the markers for the seaward and landward situated dunes, so that they can be differentiated.

o I truly enjoyed the discussion part. I checked the references and found them all correct and found them all correct (i.e. references cited in the text are in the list and vice versa). The manuscript is well written. I could not find grammatical errors or awkward sentences that would distract me as a reader. On the contrary, the text is easy to follow. I believe the manuscript should proceed to publication after the revision outlined above. UAV systems are becoming more and more accessible to a wider community and hence I believe contributions such as the one outlined in this paper will be welcome by a number of other coastal researchers.

- Thank you for your comments, we are glad you liked our discussion.

---

## Author Comment (AC2) · 2 Oct 2017

Thank you for your comments and the helpful feedback, which will help us improve both clarity and impact of the MS. Below we provide a point-by-point response to the comments, including their consequences for the MS.

Reviewer comments are indicated with open bullet points, whereas our response is indicated with a dash.

Kind regards, also on behalf of all co-authors

Marinka van Puijenbroek

This is an interesting topic , sadly very poorly written.

o Line 55- there are multiple papers outlining how incipient or embryo dunes develop in multiple countries so this is patently wrong – remove or rephrase.

- We agree that there are many paper that describe the formation of incipient foredunes or embryo dunes. However there are not so many papers that quantify the factors that determine the speed of early dune development. We will adapt the sentence to reflect this.

o Lines 57 to 63- actually Hesp stated that incipient foredunes are initiated in several ways and by nebkha and shadow dune formation is only ONE way. If the authors are going to review how incipient foredunes are formed they need to state all the other ways too – e.g. by aeolian deposition in continuous alongshore canopies of vegetation as well as discrete nebkha. And it's: incipient foredunes" NOT incipient dunes" - the latter describes any type of dune. . .

- In our study site dune formation is initiated by the establishment of vegetation and the formation of a nebkha and shadow dune. Since the formation of an incipient foredune by sand deposition within the continuous alongshore vegetation did not occur in our study site, we would rather not add this process to our introduction. We will clarify throughout our MS that we are studying nebkha dunes.

o Lines 79-80 these refs are very recent – the more comprehensive reviews of e.g. effect of veg density and distribution are in hesp papers – 1983, 1988 for example so cite these and Arens papers.

- We only cited the more recent papers to limit word counts. We will add some additional older references including Hesp from 1983 and 1988 as well as the papers by Arens, to give a more comprehensive overview.

o Lines 91-92. You need to explain better WHY u think greater dune size should mean

greater accretion/deposition. Is it because u think if a dune is big then it obviously has a greater sediment supply than a small dune? BUT what about age? How has this been taken into account? A dune might be small because its young/in early development stage, a big one because it's been sitting there for 200 years or gets regular scarping, scarp fill, crest growth due to that. . . Also is it because a larger vegetation patch would produce a larger nebkha and therefore would be able to collect more sand? There are multiple answers here and you must discuss there and later in the discussion/conclusions the impacts of these on your results.

- We changed our hypotheses to clarify our expectations:

We expected that nebka dune growth would be a function of vegetation density, initial dune size, and shelter, with the function being modulated by season and degree of shelter. We hypothesised that:

1) Nebkha dunes with high vegetation density grow fastest irrespective of season or shelter

2) In summer, growth of nebkha dunes is linearly related to initial dune size with small dunes growing at the same rate than big dunes. Exposed dunes grow faster than sheltered dunes because of higher sand supply.

3) In winter dune growth is no longer linearly related to initial dunes size, as small dunes are more susceptible to storm erosion than big dunes. Exposed dunes grow slower than sheltered dunes because of higher storm erosion.

- The dunes in our study are quite young, most of the nebkha dunes (ca. 95%) have developed within 5 years. Age is important as it will affect the size of the nebkha dunes, however age is difficult to measure. Furthermore, in coastal systems the dune size can also decrease by sea water inundation during large storms, this erosion will weaken the correlation between age and nebkha dune size. At the study site section we mention the age of our nebkha dunes.

- The area of the vegetation patch can indeed have a large effect on the sand deposition and thereby nebkha dune growth. We therefore did some additional analysis to test the effect of vegetation area on nebkha dune growth. In our study site the vegetation area was correlated to the dune size. We checked whether the vegetation area is a better predictor for nebkha dune growth than dune size, however this was not the case. Especially for the dunes seaward of the foredune, vegetation area only explained 36% of the variation, whereas dune size explains 90% of the variation. We will include these results and discuss this in the discussion.

o Lines 92-93: WHY? Because of snow cover, more wave energy and erosion, wet sand WHAT? Please explain.

- We think that exposed dunes grow faster in summer, because there is no storm erosion and therefore more net sand deposition, the sheltered dunes will grow slower because they have less sand supply. In winter storms result in sand erosion, potentially leading to negative growth for the exposed dunes. The sheltered dunes are protected from the storm and will still have a positive growth and therefore have an increased growth in winter. To clarify our expectation we will change the hypotheses, see above new version.

o Lines 101-102: WHAT 3 types of dunes? You haven't said before this that there are 3 types. In line 100 u say dunes are formed by 1, 2 or a mixture. . . is that what u mean by saying 3 TYPES of dunes? In which case they are NOT types.(im convinced even by this stage you do not understand how dunes are classified. . .) they are ALL incipient foredunes formed in diff species or mixtures of species. REWRITE. Elucidate please!

- We will change the sentence to reflect that these are all nebkha dunes, with different species composition. We will check the manuscript to clarify that we are always talking about nebkha dunes and that the different dunes consist of different plant species.

o It is NOT obvious until one gets into the methods section that you are mostly, or

entirely talking about incipient foredunes and mostly nebkha and shadow dunes. You need to state this clearly at the start of the paper and also in the abstract.

- We indeed study nebkha dunes only. We will change the text accordingly.

o Lines 275-276: dune height - WHY? Because these are older since they are more landward? Explain 289-290: obviously because they formed earlier and are older and have had a greater time to collect sand. How about stating these kinds of associations when u state your results?

- The sheltered dunes are not much older than the exposed dunes, five years at most. Nevertheless, we agree that the height differences between sheltered and exposed dunes cannot be contributed to their position only, but can be a function of their slightly older age too. We added this explanation to the MS.

o Also u are omitting the important papers on flow and sedimentation in patches or vegetation – classic study of diff patch density by Qian et al; Liu papers, Bouma paper on flow in veg patches underwater etc. – these all provide excellent explanations of how density controls nebkha development and need to be reviewed and cited.

Bouma, T.J., van Duren, L.A., Temmerman, S., Claverie, T., Blanco-Garcia, A., Ysebaert, T., Herman, P.M.J., 2007. Spatial flow and sedimentation patterns within patches of epibenthic structures: Combining field, flume and modelling experiments. Continental Shelf Research 27, 1020–1045.

Dong.,Z., Wanyin, L., Guangqiang, Q., Ping, L., 2008.Wind tunnel simulations of the three-dimensional airflow patterns around shrubs. Journal of Geophysical Research 113: F202016, doi: 10.1029/2007JF000880

- We thank you for calling attention to these nice papers; we will incorporate them into our MS, making our discussion stronger.

o Lines 337-340: it's strange and weird that you state dune vol is related to dune volume! Of course it is as it's the same thing. . . . Rewrite to explain better what you

are correlating here.

- We meant that the absolute change in dune volume was related to the initial dune volume, we will rewrite this sentence to make it more clear.

o Line 358: YOU MEAN: "The aim of this study was to explore the contributions of vegetation and dune size to NEBKHA dune development" - add this word otherwise its totally confusing and non-obvious what u are talking about; i.e. ANY dune development??!

- We changed this to nebkha dune development.

o Line 359 – now your aim is ONLY about degree of shelter? What about the other aims stated at the start of the paper??

- The main aim of our study is to explore the contributions of vegetation and dune size to nebkha dune development. Our secondary aim is to understand how the contribution of vegetation and dune size is modified by the degree of shelter. We will change the sentence to better reflect this.

o Lines 368-369: because you have failed to adequately review the literature you are stating untruths here. One of the great papers to fully show how seasons control fore-dune growth is the one by Davidson-Arnott (ref in Hesp 2002 paper maybe). Check his book which has the model in it I think. At any case remove the statement that this is the first to relate foredune growth to seasonal change.

- You are entirely correct that we are not the first paper to show how seasons control vegetated dune growth. Davidson-Arnott and Law (1990) show that the amount of sand deposition at a foredune depends on the season, where in winter more sand is deposited than in summer. Montreuil et al. (2013) showed that embryo dunes show a seasonal cycle of summer growth and winter erosion. As far as we know, we are the first paper to show that the effect of vegetation and dune size on nebkha dune development differs between a winter and summer. We will change the sentence to

clarify this.

o Lines 390-393: the referencing of the transverse dune lit here doesn't compute. Shadow dunes and/or nebkha do not at all have the same flow dynamics as transverse dunes. You need to rethink this entire idea and writing. Shadow dunes for example are controlled by paired horizontal flow vortices and max slope angle (hesp 1981). Nebkha vol and height is largely controlled by veg density and nebkha age and rate of plant growth.

- You are correct that it is not correct to compare nebkha dunes with transverse dunes. We therefore removed the sentence.

- The sentence will be replaced by the following sentence: The linear relationship between initial dune volume and dune volume change found for the nebkha dunes in our study indicates that different dune sizes have similar effect on the wind flow pattern per unit of area, which indicates scale invariance (Hallet, 1990). Scale invariance has been used for modelling nebkha and foredune development (Baas, 2002; Durán Vinent and Moore, 2013), but has not yet been validated for nebkha dunes to our knowledge.

o Lines 415-416- and less storm surge, wet high tide beach, etc on the sheltered side??

- We compared the dune growth of sheltered nebkha dunes between summer and winter. In winter the sheltered dunes had a slightly higher growth compared to summer. This higher growth rate cannot be caused by less storm surge, since these nebkha dunes were not affected by storm surge in summer and winter. Therefore, the higher growth rate is probably caused by higher sand deposition in winter. Although, at a wet high tide beach less transport is also possible, we expect that the higher wind speed in winter are a more likely explanation for the higher dune growth for sheltered dunes in winter. We will change the sentence to clarify our result, to the following: Interestingly, the sheltered dunes had a slightly higher dune growth in winter compared to summer. This increase in dune growth for sheltered dunes can perhaps be explained by more frequent and/or intensive aeolian transport events during winter resulting into higher

sand supply to the sheltered dunes.

o Line 418 and subsequent lines: You are NOT describing "veg characteristics" here. BE specific – u are at least first describing the effect of veg species differences or combinations of species, NOT density, distribution, height etc. So be specific – rewrite. OK I see that you discuss these other factors next BUT would be better to still rewrite the first part to make it clear you are first just talking about species differences. Lines 448-449: there are several studies showing that ammophila does trap more sand generally compared to other species due to its high density clump-like nature so cite some of these.

- You are correct that we first discuss the difference in dune growth formed by different plant species. To make the title better reflect the section we will rename the title to vegetation. Furthermore, we will be more specific in the subsequent lines on our results.

- We added a reference that reported that A. arenaria can trap more sand compared to other dune building species.

o lines 514-515: I don't see anywhere a decent explanation of why this is the case. You need to better explain this conclusion.

- Thank you for calling attention to this. Indeed, we only looked at the difference in dune growth for dunes with different species composition. We will change the sentence to the following: Species composition does not affect dune growth over summer, but does affect dune growth during winter, particularly at exposed sites.

---

## Referee Report (RR1)

The new version of the manuscript is much better and clearer with the changes in the text and figures. I have some minor remarks:

- Lines 198 and 427: it must be all in lower case letter for 'photogrammetric'.
- Lines 430-431 are confusing and need to be split into 2 sentences.
- Lines 529-531: the meaning of 'co-variation' is not clear in this sentence.
- Lines 550-552 must be re-written.

---

## Author Response (AR2)

**Author response on "Exploring the contributions of vegetation and dune size to early dune building using unmanned aerial vehicle (UAV)-imaging" by Marinka E. B. van Puijenbroek et al.**

I would like to thank both reviewers for their positive comments and the helpful feedback, which helped us improve both clarity and impact of the MS. In this version of our manuscript, we addressed the comments of one of the reviewers (see below), thanked the reviewers in the acknowledgements and included information about the accessibility of the final dataset (in the marked-up version of the manuscript below).

Kind regards, also on behalf of all co-authors

Marinka van Puijenbroek

Comments Anne-Lise Montreuil

The new version of the manuscript is much better and clearer with the changes in the text and figures.

I have some minor remarks:

- Lines 198 and 427: it must be all in lower case letter for 'photogrammetric'.
    - *We thank the reviewer for noticing and changed the p to a lower-case letter.*
- Lines 430-431 are confusing and need to be split into 2 sentences.
    - *We changed it into two sentences and rewrote the second part to clarify our results.*
- Lines 529-531: the meaning of 'co-variation' is not clear in this sentence.
    - *Co-variation was not the correct word, we changed it to collinearity.*
- Lines 550-552 must be re-written.
    - *We rewrote that sentence, to clarify our message.*

[revised manuscript text omitted]